# Transcriptomics of *Plasmodium vivax* rosetting

**Catarina Bourgard[1], Julia Weber Ferraboli[2], Stefanie Costa Pinto Lopes[3,4],
Marcus Vinicius Guimarães de Lacerda[3,4], Per Sunnerhagen[5], Letusa Albrecht[2]/+,
Fabio Trindade Maranhão Costa[1]/+**

[1]Universidade de Campinas, Departamento de Genética, Evolução, Microbiologia e Imunologia, Laboratório de Doenças Tropicais Prof Dr Luiz Jacintho da Silva, Campinas, SP, Brasil
[2]Fundação Oswaldo Cruz-Fiocruz, Instituto Carlos Chagas, Curitiba, PR, Brasil
[3]Fundação Oswaldo Cruz-Fiocruz, Instituto Leônidas e Maria Deane, Manaus, AM, Brasil
[4]Fundação de Medicina Tropical Dr Heitor Vieira Dourado, Gerência de Malária, Manaus, AM, Brasil
[5]University of Gothenburg, Department of Chemistry and Molecular Biology, Sweden

**BACKGROUND** The mechanisms that underlie the pathobiology of the neglected malaria parasite *Plasmodium vivax* are poorly known. It has been demonstrated that *P. vivax* has the capacity to remodel the infected host reticulocyte membrane and promote adhesion, which might play a role in host immune evasion. *P. vivax*-infected red blood cells (*Pv*-iRBCs) bind to the host endothelium, fibroblasts in the spleen and normocytes, rather than reticulocytes. The adhesion of Pv-iRBC to at least two red blood cells is known as rosetting. Rosetting is frequently observed in *P. vivax* isolates, yet there is limited knowledge about the specific molecular players involved in this phenomenon. Throughout the development of the parasite in the bloodstream, all stages, including gametocytes, have the ability to form rosettes.

**OBJECTIVES** In this study, we evaluated the molecular basis behind rosetting phenotypes.

**METHODS** We employed *P. vivax* rosetting *ex vivo* assays followed by RNA-seq, to sequence the whole transcriptome of parasite populations with distinct rosetting characteristics.

**FINDINGS** Among the 492 differentially expressed genes of *P. vivax* isolates with high rosetting (HR) versus low rosetting (LR) formation, 172 (34,96%) are annotated as genes conserved within *Plasmodium* and of unknown function. The expression profiles of the other 320 genes (65,04%) highlight the importance of integral membrane proteins and membrane-associated proteins with adhesive or adhesin-like properties, representing 10% of the transcribed genes (53 genes), such as *Plasmodium* Helical Interspersed Sub-telomeric (PHIST) proteins in rosetting phenotypes. Transcriptomic analyses revealed that approximately 4% (19 genes) of differentially expressed genes were kinases and 50% (248 genes) other proteins. Among cell surface proteins and integral/membrane-associated proteins, differentiated expression and positive regulation of representative 6-cysteine gene family were observed in HR formation group, which includes a tryptophan-rich protein (TRAG16), the 41K blood stage antigen precursor 41-3 protein, and merozoite surface protein 7-like (MSP7-like).

**MAIN CONCLUSIONS** These results contribute to understanding the molecular basis of *P. vivax* rosetting.

Key words: *Plasmodium* vivax - rosetting - transcriptome

*Plasmodium vivax* is the most widespread malaria parasite outside Sub-Saharan Africa, placing billions of people at risk of infection, and imposing major health and economic burdens.[1] Infection occurs in genetically distinct populations with heterogeneous resistance to chloroquine.[2,3,4,5] Severe clinical complications, although scarce,[6] have been of great concern.

Clinical complications of malaria caused by *Plasmodium falciparum* is often associated with parasite adhesive features such as the capacity of *P. falciparum*-infected red blood cell (Pf-iRBCs) to form rosettes. A rosette is characterised by the adhesion of an infected red blood cell (iRBCs) to cytoadhere to two or more non-infected red blood cells (RBCs), mediated by surface ligands that interact with surface receptors.[7] The phenomenon of rosette formation was first discovered in *Plasmodium fragile*, in the study of non-human primates[8] and later reported in *P. falciparum*.[9] Thus, the interactions involved in rosette formation are better understood in *P. falciparum*, partly due to a well-established *in vitro* culture for this species.[10]

In *P. falciparum*, three families of variant surface antigens are best known for promoting this adherence: *P. falciparum* erythrocyte membrane protein 1 (PfEMP1),[11] repetitive interspersed family (RIFIN),[12] and subtelomeric variant open reading frame protein (STEVOR).[13] Within the three families, PfEMP1s are the most studied and characterised, binding to comple-

Financial support: FAPESP (grants 2012/16525-2 and 2017/18611-7), CNPq (Universal 431403/2016-3).
CB was supported by FAPESP PhD fellowship no. 2013/20509-5; FTMC and MVGL are CNPq research fellows.
+ Corresponding authors: fabiotmc72@gmail.com | ⓘ https://orcid.org/0000-0001-9969-7300 / letusa.albrecht@fiocruz.br | ⓘ https://orcid.org/0000-0001-6406-2057

**Handling editor:** Cláudio Tadeu Daniel-Ribeiro | ⓘ https://orcid.org/0000-0001-9075-1470

ment receptor 1 (CR1),[14] heparan sulphate, and trisaccharides of blood groups A and B[15,16] — this interaction being the most understood. In the case of RIFINS, it is speculated that the binding occurs through interaction with the glycophorin A receptor and blood group A trisaccharides,[12] while STEVORS use only glycophorin C to mediate the adhesive phenomenon.[13]

In contrast to *P. falciparum*, little is known about rosetting in *P. vivax*. In fact, the relation between *P. vivax* rosetting, disease severity, parasitaemia, and blood type is still unknown.[17,18,19] Nevertheless, in *ex vivo* experimental assays, the need to use trypsin to disrupt rosettes suggests the existence of a parasitic protein(s) involved in the process of *P. vivax* rosette formation.[18] Moreover, the incidence of rosetting observed in *P. vivax* isolates is more accentuated than in falciparum malaria, whereas asexual and sexual stage parasites can form rosettes.[20] The rosette complex structure is based preferentially on mature erythrocytes (normocytes), whereas glycophorin C acts as a red blood cell receptor for vivax rosetting.[20] *P. vivax*-infected red blood cells (*Pv*-iRBCs) have altered rheological properties, principally in the decrease of membrane elasticity, which enables them to avoid splenic clearance. According to studies on deformability of *Pv*-iRBCs, rosette-forming *Pv*-iRBCs are distinctly more rigid than their non-rosetting counterparts, indicating that rosette formation by schizonts *Pv*-iRBCs contributes to parasite retention in the host microvasculature and/or spleen.[21]

Because of the development of single-cell RNA sequencing (scRNA-seq), gene expression throughout the life cycle of *P. vivax* has been characterised and shown that genes are transcribed into multiple isoforms, which is regulated by parasitic development.[22] Comparative transcriptomics also reveals differential gene expression between geographic isolates of *P. vivax* and implications for erythrocyte invasion mechanisms.[23]

Transcriptomic studies have been used to better understand parasite-host interactions. It has been found that *P. vivax* invasion mediated by band 3 is associated with transcriptional variation in *P. vivax* tryptophan-rich antigen genes (*Pv*TRAg38).[24] Transcriptomics was also used to estimate the proportions of each stage of the parasite's life cycle present in mass RNA sequencing data, using the gene expression deconvolution strategy from a reference database.[25] Finally, RNAseq data suggest that genes from the multigene family of plasmodial interspersed repeats (*pir*) show geographically conserved transcription are also present throughout the mosquito life cycle.[26]

Here, our efforts concentrate on understanding the expression patterns that could reveal which parasite ligands or metabolic pathways might be involved in the reshaping of normocyte cell membranes and properties that are related to *P. vivax* rosetting capacity.

### SUBJECTS AND METHODS

*Ethical approval* - All methods were performed in accordance with the relevant guidelines and regulations. All procedures, including protocols and consent forms, were approved by the Ethics Review Board of Fundação de Medicina Tropical Heitor Vieira Dourado (FMT-HVD), a tertiary care centre for infectious disease in Manaus, Amazonas State, Brazil (process CAAE-0044.0.114.000-11 and 54234216.0000.0005). Informed consent was sought and granted by all patients.

*Study area, subjects and sample collection* - Vivax malaria patient recruitment was made at FMT-FVD. Adult patients diagnosed with uncomplicated *P. vivax* malaria were recruited, whereas severe malaria, patients under anti-malarial treatment, with *P. falciparum* malaria and/or *P. falciparum* and *P. vivax* mixed infections and pregnant women were excluded [Supplementary data (Table I)]. Conventional thick-smear microscopic diagnosis of *P. vivax* malaria and parasitaemia determination were done before initiation of treatment, when up to 8 mL of peripheral blood was collected from each patient in citrate-coated Vacutainer™ tubes (Becton-Dickinson) and *P. vivax* mono-infection was confirmed by polymerase chain reaction (PCR) analysis, as described elsewhere.[27]

*Parasite isolation, enrichment and ex vivo maturation* - To obtain enriched *Pv*-iRBCs, samples were immediately processed. After sample centrifugation at 400 x g for 5 min at room temperature, plasma and buffy coat layer were removed and the pellet was resuspended in an equal volume of RPMI parasite medium (McCoy-5A, Gibco) and then CF11 column filtration (Sigma) was performed to deplete white blood cells (WBC).[18,28,29] Before, during, and/or after short *ex vivo* culture, thin blood smears were prepared and stained with *Panótico Rápido* (Laborclin) kit to control the parasite maturation. According to the stages of parasite maturation of each sample, the early blood staged parasites were cultured for 18 to 22 hours to allow them to mature until late trophozoites and/or schizonts as follows: 5% haematocrit in McCoy-5A medium supplemented with 20% of human AB serum, incubated at 37°C with a gas mixture containing 5% $CO_2$, 5% $O_2$, 90% $N_2$.[18,30] Afterwards, parasite enrichment was done through Percoll 45% gradient protocol as previously described.[31]

All samples were processed immediately after collection following the procedures for parasite isolation and enrichment to obtain parasitaemia > 50%, and thus, a total number of *Pv*-iRBCs greater than 400,000 to enable us to proceed with rosetting assays. Thin blood smears after Percoll 45% enrichment allowed us to choose isolates with a higher percentage of trophozoite to schizont staged parasites and control for host lymphocyte contamination.

*Rosetting rates assessment* - A rosetting methods scheme to determine the rosetting capacity of distinct populations of *Pv*-iRBCs from *vivax* malaria patient isolates[20,31] is shown on [Supplementary data (Fig. 1)]. In brief, 20 µL of *Pv*-iRBCs at 2.5 to 5% parasitaemia and at 4% haematocrit were incubated for 40 min at 37°C in rosetting media (McCoy's 5A medium supplemented with 20% of patient autologous plasma). Triplicate aliquots of each sample were stained with 45 µg.ml⁻¹ of acridine orange and examined by direct light and fluorescence microscopy (Nikon Eclipse 50i, filter 96311 B-2E/C). Rosetting rates were accessed by counting

approximately 200 *Pv*-iRBCs in triplicate. The rosette complex was defined by the binding of two or more uninfected erythrocytes to a *Pv*-iRBC.[20,31]

*Plasmdium falciparum in vitro cultures - P. falciparum* FCR3 S1.2[32] was cultured according to standard procedures previously described.[33] In summary, *P. falciparum* FCR3 S1.2 strain was cultured in purified erythrocytes from O⁺ healthy local donors in RPMI 1640 (Gibco) supplemented with 5% AlbuMAX (Gibco), sodium bicarbonate (25 mM; Sigma), hypoxanthine (100 µM; Sigma) and gentamycin (50 µg.L$^{-1}$; Gibco). *Pf*-iRBC*s* were maintained at 2% haematocrit and incubated at 37ºC with a gas mixture containing 5% $CO_2$, 5% $O_2$, 90% $N_2$. Cultures were synchronised with a 5% sorbitol solution and further enriched for mature stages in 60% Percoll. Thin blood smears were prepared and stained with *Panótico Rápido* (Laborclin) kit during *in vivo* culture to control the parasite maturation and parasitaemia.

*RNA extraction and quality control, low input cDNA synthesis and library preparation for RNA-seq* - As previously published,[31,34] we followed the same RNA preparation methodology suitable for RNA-seq harvested from *P. vivax*-infected patients. To summarise, total RNA extractions from all samples were accomplished by using the RNeasy Micro kit (Qiagen) according to the manufacturer instructions. Quality control was done by electrophoresis of the extracted RNA samples in the Agilent 2100 Bioanalyzer instrument using the Agilent RNA 6000 Pico Kit reagents and chips and analysed on the 2100 Expert software, according to the Agilent Technologies recommendations [Supplementary data (Table II)]. SMART-Seq V4 Ultra Low Input RNA kit was used for cDNA libraries generation. cDNA quality, quantity and size range were evaluated through Bioanalyzer using the Agilent High Sensitivity DNA Kit (cDNA, 5 to 500 pg.µL$^{-1}$ within a size range of 50 to 7000 bp), as per manufacturer instructions [Supplementary data (Table III)]. The Covaris AFA system was used for controlled cDNA shearing, resulting in DNA fragments between 200 and 500 bp sizes. Instructions were followed as indicated in the SMART-Seq V4 Ultra Low Input RNA kit for sequencing user manual by Clontech Laboratories, Inc. A Takara Bio Company. cDNA output was then converted into sequencing templates suitable for cluster generation and high-throughput sequencing through the Low Input Library Prep v2 (Clontech Laboratories, Inc. A Takara Bio Company). Library quantification procedures using the Library Quantification kit (Clontech Laboratories, Inc. A Takara Bio Company) by the golden standard qPCR and Agilent's High Sensitivity DNA kit were successfully completed [Supplementary data (Table IV)], before proceeding for the pool set-up at a final concentration of 2 nM for direct sequencing. The generated libraries were cluster amplified and sequenced on the Illumina platform using standard Illumina reagents and protocols for multiplexed libraries and by following their loading recommendations. Sequencing runs were performed on HiSeq 2500 sequencer on Rapid Run mode with the HiSeq Rapid Cluster Kit v2 (100x100) Paired End, HiSeq Rapid SBS Kit v2 (200 cycles) and HiSeq Rapid Duo cBot v2 Sample Loading kits from Illumina, Inc. Work samples were selected based on the quality and quantity of samples after the extraction step, given the difficulty of obtaining quality materials for RNA-seq assay. For this reason, four patients with low rosette (LR) formation rates (< 10%) and two patients with high rosette (HR) formation rates (> 10%) were selected.

*Raw reads alignment and mapping* - The sequence data was analysed using the EuPathDB-Galaxy hub (https://eupathdb.globusgenomics.org/) free, interactive, web-based platform for large-scale data analysis by assembling a new workflow adapted for the RNA-seq experimental design on Galaxy platform[35] and using the PlasmoDB[36] pre-loaded *P. vivax* reference genome.[37] In summary, raw reads were checked for quality by running Fast Quality Control (Galaxy Tool Version FASTQC: 0.11.3; https://www.bioinformatics.babraham.ac.uk/projectY/fastqc/), a java quality control tool for high throughput sequencing data. Illumina adaptors were trimmed through Trimmomatic (Galaxy Tool Version 0.36.5) on the Illumina paired-end data[38] [Supplementary data (Fig. 2, Table V)], and read alignment and mapping was performed with TopHat2[39] (Galaxy Tool Version SAMTOOLS: 1.2; BOWTIE2: 2.1.0; TOPHAT2: 2.0.14), towards the *P. vivax* P01 reference genome from PlasmoDB release 38 known transcripts and splice junctions [Supplementary data (Table VI)]. FPKM estimation by count of the number of aligned reads matching the annotated reference genes was executed with htseq-count to FPKM[40] (Galaxy Tool v. HTSEQ: default; SAMTOOLS: 1.2; PICARD: 1.134). Final differential gene expression was performed with DESeq2[41] Galaxy Tool (v. 2.1.6.0). Differential gene expression between the different analysis groups was identified after a pairwise Wilcoxon test was used to compare the transcriptional profiles with the following cutoffs: q-value < 0.05 and a log2 fold change > 1.5. From the final list of differentially expressed genes identified, we selected those that could be informative about the host expression differences between high against low rosetting *P. vivax* infecting populations, focusing on the relative presence or absence within the sample comparison.

## RESULTS

*Evaluation of rosetting in Brazilian P. vivax isolates* - The rosetting capacity was evaluated in 16 isolates of *P. vivax* collected from infected patients in Manaus, Brazilian Amazon. Samples were mainly from male individuals (2.2 M:F ratio) with an average of 39.5 years old and presented a range from 4,800 to 19,200 parasites per µL of blood [Supplementary data (Table I)]. All patients were thrombocytopenic (platelet levels <150,000 platelets per microliter of blood) [Supplementary data (Table I)]. Rosetting rates varied from 4% to 56.88% and there was no relation with blood parameters [Supplementary data (Table I)]. Isolates with less than 10% rosettes were considered LR while isolates with more than 10% rosetting rates were considered HR (Fig. 1). There was a significant difference in rosetting rates between LR and HR groups (Mann Whitney test p < 0.001, Fig. 1). The prevalence of

the parasite stage for each isolate is indicated separately according to the groups analysis, with mature forms in HR group and young forms in LR (Table).

*Plasmodium vivax* low *input cDNA synthesis, library preparation and sequencing* - To evaluate the expression profile of *P. vivax* parasites with LR or HR features, parasite RNA extractions were done using the RNeasy® Micro kit and its quantity and quality evaluated using the Bioanalyzer® platform [Supplementary data (Table II)]. Only six of the 16 patients had samples selected for transcriptomics assays due to the sample quality after RNA extraction. On average 11.33 pg/μL of RNA, ranging from 1.87 to 21.5 pg/μL, was obtained with an average RNA integrity number (RIN) of 8.0 (ranging from 6.3 to 9.8). Given the low amounts of *P. vivax*

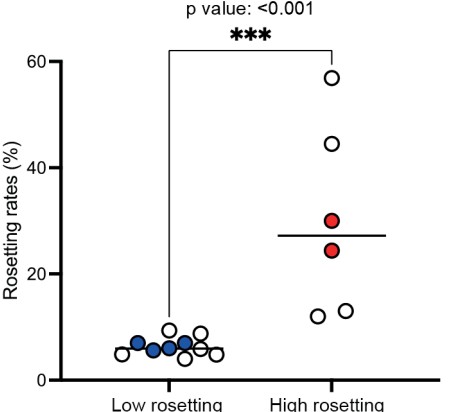

Fig. 1: rosetting of *Plasmodium vivax*. *P. vivax*-infected red blood cells (Pv-iRBC) rosetting was evaluated in *P. vivax* isolates from the Brazilian Amazon. A rosetting was identified when an infected cell was adherent to at least two non-infected ones. Isolates with more than 10% rosetting rates were considered high rosetting. Circles in red indicate high rosetting isolates selected for transcriptomics and blue circles indicate low rosetting *P. vivax* isolates selected for transcriptomics analysis. The line indicates the rosetting cut off point. The non-filled circles represent *P. vivax* isolates that were not included in transcriptomics analysis due to the quality of the material after RNA extraction. A Mann Whitney test p < 0.001 was used for statistical analyses to differentiate in rosetting rates between low and high rosetting groups.

RNA, we opted to use the SMART® technology, which offers unparalleled sensitivity and unbiased amplification of cDNA transcripts from low input RNA samples. Immediately thereafter, the cDNA output was converted into sequencing templates suitable for cluster generation and high-throughput sequencing resulting in a sequencing-ready library for the Illumina® platform [Supplementary data (Tables III-IV)].

*Whole transcriptome shotgun sequencing data analysis* - We obtained a total number of 2,638,465 raw reads [Supplementary data (Tables V-VI)]. On average 439,744 paired end reads (100 bp) were obtained per sample from the 6 libraries that were successfully sequenced, with an average GC content of 46.6% [Supplementary data (Tables V-VII)]. The total number of FastQC raw reads obtained from all libraries revealed good sequence quality and trimming steps whereas only a minor percentage of reads (2.1%) were excluded [Supplementary data (Table V)], mainly repetitive, not accurately determined, and Illumina® adaptors run through sequences. Using the *P. vivax* P01 reference genome, we were able to align and map the total number of trimmed reads obtained to annotated protein-coding genes [Supplementary data (Table VI)]. Sequences showing multiple or discordant alignments were excluded from the analysis.

*Expression profiles associated to P. vivax rosetting* - The transcriptomes of rosetting parasites were evaluated in isolates with HR (69U15 and 106U16) and LR capacity (63U15, 65U15, 73U15 and 109U15) (Fig. 1). Through RNA-seq data analysis, we accessed the differential gene expression profiles between these two groups, and dissected by data mining possible differences that might explain *P. vivax* rosetting during the progress of vivax malaria disease. Analysis of HR versus LR *P. vivax* isolates revealed a group of 492 differential expressed genes [q-value < 0.05, Fig. 2 and Supplementary data (Table VIII)]. Among those genes, 172 were annotated as conserved *Plasmodium* spp. proteins of unknown function (34.96%) (Fig. 3A). The remaining 320 genes were grouped in cell surface and integral membrane/ membrane-associated proteins (53 genes), kinases (19 genes), and other proteins (248 genes). From the pool of

TABLE

The prevalence of the parasite stage for each isolate is indicated separately according to the groups analysis for the six samples that were selected to proceed with transcriptomic testing. Isolates with more than 10% rosetting rates were considered high rosetting while isolates with less than 10% rosettes were considered low rosetting. It indicates the sample code, smear stages after Percoll 45% (hour post invasion - hpi) and frequency of rosetting formation (%)

| Sample code | Smear stages after Percoll 45% (h) | Rosetting (%) |
|---|---|---|
| 106U16 | mature trophozoites (15-22hpi) | 30.00 |
| 69U15 | Schizonts (21-26hpi) and some merozoites | 24.40 |
| 63U15 | young trophozoites (< 11-13 hpi) | 7.00 |
| 65U15 | young trophozoites (12-15 hpi) | 7.00 |
| 73U15 | Some rings, young trophozoites (11-15 hpi) and some gametocytes | 5.60 |
| 109U16 | rings and young trophozoites (11-17 hpi) | 6.00 |

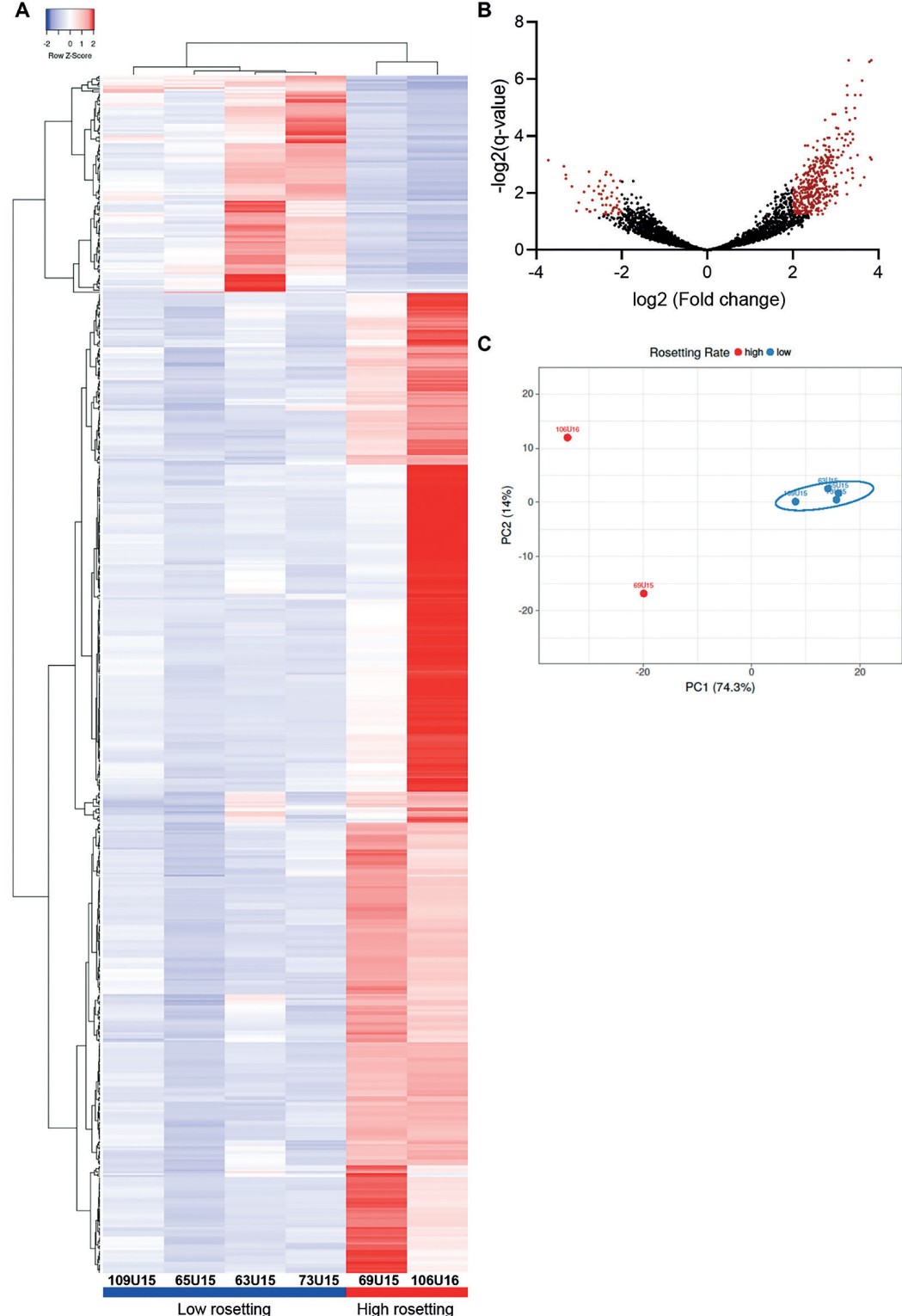

Fig. 2: heatmap clustering (A) of differentially expressed genes between isolates registering low (63U15, 65U15, 73U15 and 109U15) versus high rosetting rates (69U15 and 106U16), using complete linkage hierarchical clustering method and Pearson's distance measurement method for computing distance between rows and columns.[69] Volcano plot (B) showing the range of the log2 (Fold change) relative to the -log2(q-values) of *Plasmodium vivax* mapped genes. Identified genes with a q-value < 0.05 and -2 < log2 (Fold change) > 2 cut-offs obtained from RNAseq differential gene expression analysis are put in evidence (dark red dots). PCA analysis (C) with unit variance scaling applied to rows, SVD with imputation is used to calculate principal components. X and Y axis show principal component 1 and principal component 2 that explain 74.3% and 14% of the total variance, respectively. N = 6 data points, 69U15 and 106U16 isolates showing higher rosetting rates (red dots), and 63U15, 65U15, 73U15 and 109U15 isolates yielding low rosetting rates (blue dots; blue ellipse). Prediction ellipses are such that with probability 0.95, a new observation from the same group will fall inside the ellipse.[70]

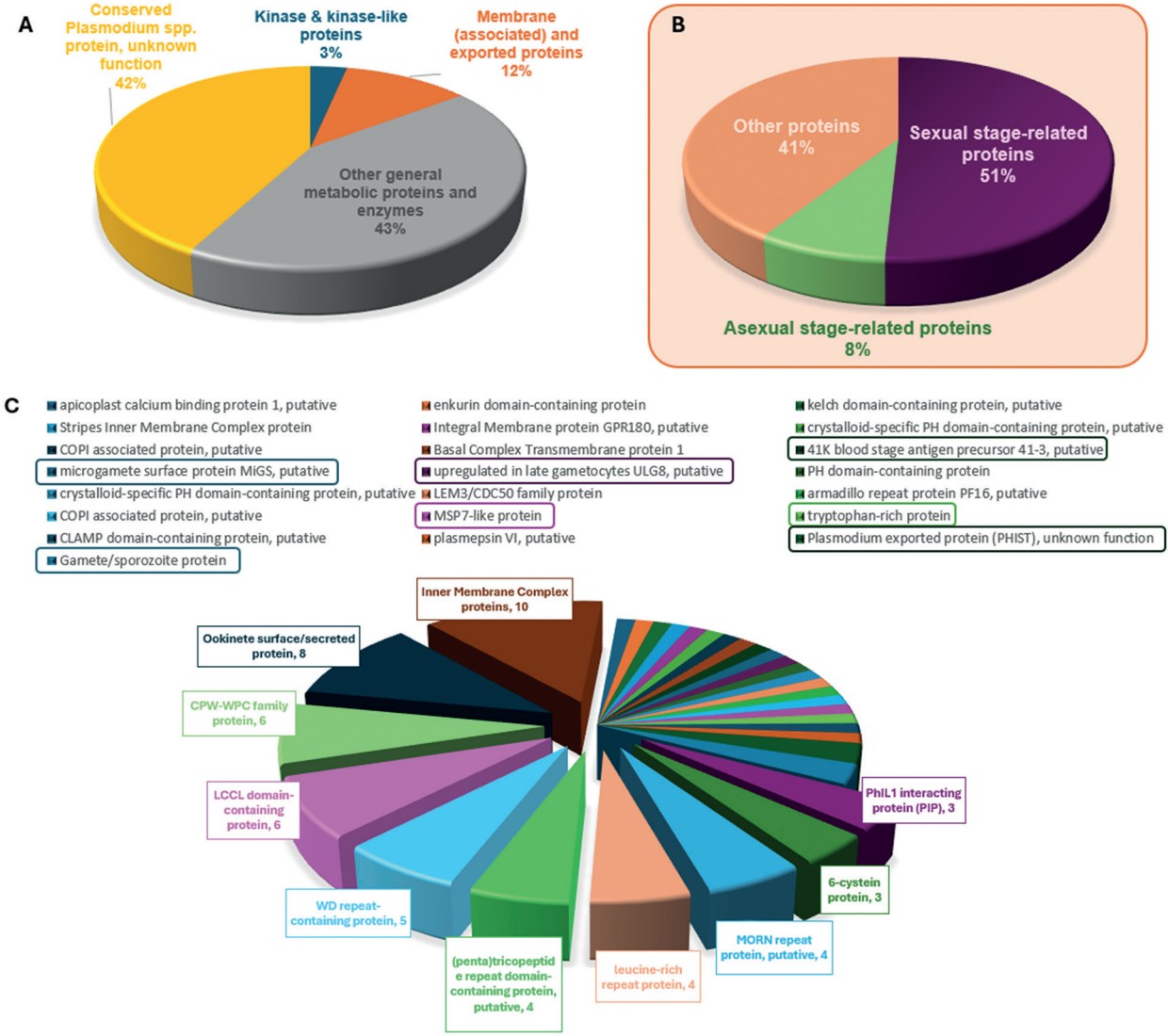

Fig. 3: pie charts showing the 492 differentially expressed genes (q-value < 0.05) grouped by protein function (A) in membrane/membrane-associated proteins (53 genes; orange), kinase enzymes (19 genes; blue), conserved *Plasmodium* spp. proteins of unknown function (172 genes; yellow) and other proteins (248 genes; grey). (B) From the pool of membrane/membrane-associated proteins (pie chart inside light-shaded orange box), proteins were found to be commonly expressed in sexual stages of the parasite (purple pie slice) and other proteins are ubiquitously expressed by *P. vivax*. The last group included proteins present in core membrane complexes (light orange) and other proteins (green) characteristically expressed in the parasite asexual stages. (C) Donut chart showing the membrane/membrane-associated proteins expressed on asexual stages including *Plasmodium* helical interspersed sub-telomeric proteins (PHIST), one tryptophan-rich protein (TRAG16), the 41K blood stage antigen precursor 41-3 protein, the merozoite surface protein 7-like (MSP7-like), and other membrane complex proteins including those characteristically expressed on parasite sexual stages [see Supplementary data (Table VIII) for further details].

cell surface and integral membrane/membrane-associated proteins (Fig. 3B), representatives of the 6-cysteine gene family were differentially expressed and upregulated in the HR group. Differentially expressed genes included one tryptophan-rich protein (TRAG16), the 41K blood stage antigen precursor 41-3 protein and the merozoite surface protein 7-like (MSP7-like) [Fig. 3C and Supplementary data (Table VIII)]. The most highly expressed genes for each patient group were listed, with positive Log2 Fold Change values corresponding to the HR group and negative values corresponding to the LR group [Supplementary data (Table VIII)].

## DISCUSSION

Rosetting is a common adhesive phenomenon observed in *P. vivax* infection. Even if this feature was first described decades ago,[17] the molecular basis of rosetting in vivax malaria is still unknown.[42] The frequency of rosetting in *P. vivax* isolates is quite high, to the extent that in some studies all isolates are forming rosettes at some degree.[20,31] However, there is currently no consensus about the ligands and receptors involved in rosetting. While glycophorin C is known to be an important receptor in rosetting,[20] no ligands have yet been described.

Comparatively with *P. vivax*, rosette formation in *P. falciparum* is a less frequent adhesive phenomenon, but it is linked to disease severity.[43] Rosette formation can be classified into three types in falciparum malaria: type I – formation mediated by parasite proteins expressed on the surface of the infected cell and receptors present in the erythrocyte; type II – in addition to the parasitic molecules and erythrocyte receptor present, host plasma factors are important; type III – secreted parasite proteins mediate the interaction between healthy erythrocytes.[44] The parasite prefers to form rosettes in blood groups A and B, generating larger rosettes compared to group O,[45,46] since the antigens A and B are indicated to act as co-receptors in *P. falciparum*.[47]

A model parasites for rosette formation in *P. falciparum* (FCR3S1.2) revealed a dominant transcript, the *var*2 gene (IT4var60), belonging to the PfEMP1 surface protein family, suggesting that it is responsible for encoding the rosette formation phenotype of the strain.[48] Furthermore, whole transcriptome analyses aimed at detecting genes related to cytoadherence revealed that another gene belonging to the PfEMP1 family showed significant positive transcriptional regulation of *var*9 in *P. falciparum* R29 strains, a clone derived from the IT/FCR3 strain.[49] Despite that, information on rosette formation obtained from experiments conducted with *P. falciparum* cannot be extrapolated to *P. vivax*. Comparison between species is problematic because the proteins involved in rosette formation in *P. falciparum* are part of multigene families that have no orthologs in *P. vivax*.[20]

Genes expressed specifically at different stages of *P. vivax* life cycle have been identified by different studies using transcriptomic tools.[50] Understanding which genes are positively and negatively regulated in the hepatic stages of the parasite allows us to find a therapeutic target against infection, since it is possible to differentiate between replicating schizonts and hypnozoites at the transcriptional level.[51] Furthermore, transcriptomic signatures of *P. vivax* sporozoites provide important information about their development and lay the foundation for a cell atlas, with genes that are conserved and unique to the species.[51]

Through RNA-seq of *P. vivax* iRBCs, we evaluated the differential gene expression profiles between two groups with different rosetting capacity (high against low) and identified expressed genes that could explain *P. vivax* rosetting during the progress of vivax malaria disease. Considering the confounding expression variability expected among different *P. vivax* clinical isolates, the comparison between the LR and LR samples revealed a group of 492 differentially expressed genes.

As anticipated, a large number of those differentially expressed genes have not yet been characterised for their protein function, but the conservation of their sequence throughout *Plasmodium* spp. may be indicative of their importance on parasite rosetting phenotype. Most conserved *Plasmodium* genes showed high upregulated expression, suggesting their involvement in molecular processes important for erythrocyte binding. The functional characterisation of these proteins should further elucidate this possibility.

Within the group of differentially expressed enzymes, 3 upregulated kinases have caught our attention, since kinases are classic targets for discovery of new therapeutic drugs. Serine/threonine-Protein Kinase (NEK3) has been reported as essential for mitosis progression in *Plasmodium berghei* blood-stage development,[52] Mitogen-Activated Protein Kinase 2 (MAPK2) seems to play an important role in stress response in *Toxoplasma gondii*,[53] and Raf Kinase Inhibitory Protein (RKIP) affects activity of another kinase, the calcium-dependent protein kinase 1,[54] which regulates several important metabolic processes reliant on calcium in *Plasmodium* spp. Studies on calcium homeostasis have reported that parasitised RBCs show an increased influx of calcium when compared to the decreased efflux of unparasitised RBCs. Calcium has been localised in the *Plasmodium* spp. compartment. For the RBC invasion by the merozoite, extracellular calcium is needed, as well as for the subsequent parasite development and maturation inside of the erythrocyte.[55,56]

*Plasmodium vivax* infections are often characterised by asynchronous populations of parasites in different stages of development and/or maturation. Although our samples were chosen with the aim to access the transcriptomic profiles of trophozoites and/or early schizont parasites, we could also catch the expression of some interesting gametocyte membrane surface genes, reflecting the importance of the study of mechanisms of *P. vivax* transmission. Together with the P48/45 surface protein,[57] P47 is one such protein, having been reported as required for optimal fertilisation in *P. berghei* and for evasion of the mosquito immune response, showing a strong signature of natural selection and population structure in the *P. falciparum* and *P. vivax* genomes.[58]

Also, we verified differential expression of two LCCL lectin domain adhesive-like proteins (LAPs), a family of conserved six modular proteins, present throughout the apicomplexan genus, which are expressed in sexual stages of *Plasmodium* parasites and reported to be involved in the formation of protein complexes required for successful *P. berghei* sporogony.[59,60]

One of the most important aspects of erythrocyte infection by *P. vivax* is the dramatic structural reorganisation of the erythrocyte membrane, driven by a network of microtubules (MT) sustained by the inner membrane complex (IMC). As expected, genes codifying the actin and tubulin backbone molecules of MTs were found to be upregulated in the HR isolates, together with a group of MT motor enzymes, IMC proteins such as Photosensitised 5-[125I] Iodonaphthalene-1-azide Labelled protein-1 (PhIL1 - PIP2 and 3-) integrating proteins, which are critical in various processes such as signal transduction and intracellular and membrane trafficking.[61,62]

Furthermore, we were able to catch the overexpression of an early-transcribed membrane protein (ETRAMP). ETRAMPs are important proteins present on the membrane of intracellular parasites of *Plasmodium* species, formed during erythrocyte invasion as an invagination of the iE cell surface during the asexual blood stage parasites. Recent studies showed that ETRAMPs have been localised on the intracellular membranes of immature schizont and at the apical organelles of newly *formed*

*P. vivax* merozoites of mature schizont and have the capacity to elicit high antibody titres capable of recognising parasites of vivax malaria patients.[63]

Together with three other expressed genes from the CPW-WPC surface protein family, another membrane protein found differentially expressed in our study was the *P. vivax* MSP7-like. Merozoite surface proteins belong to families of proteins often involved in complex *Plasmodium* invasion processes. *Pf*-MSP7 interactions with host P-selectin receptors have been demonstrated,[64] which in consequence block interactions between host P-selectin and leukocyte ligands and could underlie the mechanism for the known immunomodulatory effects of both MSP7 and P-selectin in malaria infection models. Although MSP7 in *P. vivax* has not yet been functionally characterised, there is evidence this protein is under selection and thus, being functionally important in *P. vivax*[65] and it was also found in extracellular vesicles derived from plasma of infected *P. vivax* patients.[66]

In addition, four glideosome-associated proteins are observed to be differentially expressed in the high rosetting parasites. The capacity to bind, reorient and invade new host cells is mainly powered by the "glideosome" proteins.[67] The glideosome is a macromolecular complex comprising proteins with adhesive properties. These proteins are released apically on the parasite membrane and translocated to the opposite pole of the parasite through the actomyosin system anchored in the IMC.

Finally, we have detected the differential expression of *P. vivax* macrophage migration inhibitory factor (MIF) gene. It has been reported that expressed Pf-MIF protein localises to the Maurer's cleft during asexual blood stage parasites. Pf-MIF *in vitro* treatment of human monocytes inhibited their random migration and reduced the surface expression of toll like receptor (TLR) 2, TLR4 and CD86, indicating that its release potentially modulates the host monocytes functions during acute *Plasmodium* infection.[68] In accordance with this data, our analysis performed in isolates from non-severe vivax malaria patients reported a significant downregulated expression of the *P. vivax mif* gene in parasite populations showing rosetting phenotypes.

In conclusion, taken together, these results point out the importance of integral membrane and membrane associate proteins in rosetting phenotype. Functional assays might further clarify if these proteins allow the parasites to adhere to the surface of host cells, such as healthy erythrocytes, maintaining them anchored in order to create the characteristic rosette of surrounding erythrocytes, which enable the parasite to evade from the host immune system.

## ACKNOWLEDGEMENTS

To the people that agreed to participate in this study, to the field team in the Fundação de Medicina Tropical Dr Heitor Vieira Dourado (FMT-HVD) in Manaus-AM and to the sequencing facility team from USP-ESALQ Piracicaba-SP.

## AUTHORS' CONTRIBUTION

All authors conceived the experiments. CB did the field sample collection and enrichment and rosetting assays, designed and executed the RNA-seq experiments and data analysis; CB, JWF and LA went through data mining of the obtained results and drafted the manuscript. All authors reviewed the manuscript. The authors declare no competing interests.

## DATA AVAILABILITY

All data generated or analysed during this study are included in this published article (and its Supplementary data). Deep sequencing data was deposited in Array Express, accession number E-MTAB-8385.

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

# OPEN PEER REVIEW

Memórias do IOC thanks the anonymous reviewers for their contribution to the peer review of this work.

### FIRST REVIEW ROUND

REVIEWERS' COMMENTS

### REVIEWER #1

This manuscript investigates the transcriptional differences between P. vivax isolates with low rosetting and P. vivax isolates with high rosetting.

The data presented is quite descriptive, since it does not point to a clear rosetting mechanism. However, it will be usefull for the community to advance studies with this goal.

Prior to genetic material isolation, the parasites were cultured for maturation so the same stage of each isolate was analysed. However, P. vivax maturation vary a lot, with some isolates progressing to late troph/schizonts quite fast, while other seem to arrest in earlier stages in vitro. Also, some isolates when cultured seem to have a higher than expected preference for gametocytogenesis.

Therefore, it would be helpfull for future interpretation of the data if the authors add a table with the information of % of each parasite stage, for each isolate, at the moment of sample processing for transcriptomics.

### REVIEWER #2

The manuscript addresses important aspects of the molecular mechanisms underlying the resetting of P. vivax–infected red blood cells, a topic that remains poorly characterized in P. vivax biology. The study presents original findings, and the results are described in a clear and comprehensive manner, making the work accessible even to readers who are not specialists in transcriptomic analysis. The paper is short but informative, with all transcriptomic analysis presented in tables and supplementary figures.  It´s the first work that uses transcriptomics to understand P. vivax resetting. Even with a low number of samples/ libraries analyzed (6 libraires) it provides important new information to the field.

Overall, the manuscript represents a valuable contribution to the field. I recommend acceptance for publication pending minor revisions, as outlined below.

1. Abstract:

In the abstract, the authors report that one third of the genes differentially expressed in P. vivax rosetting parasites were "conserved within Plasmodium and of unknown function." This statement, however, lacks sufficient context. The total number of differentially expressed genes must be provided; without it, the proportion cited cannot be properly interpreted or considered scientifically meaningful.

2. Introduction

In the Introduction/Background section, the authors provide a comprehensive overview of transcriptomic studies conducted on P. vivax samples. However, they must also include a discussion of the molecular mechanisms of rosetting in P. falciparum, a parasite for which this phenomenon has been more extensively studied. Incorporating this information would provide essential context for readers, enabling a more meaningful comparison and a clearer understanding of the differences in pathogenic mechanisms across malaria-causing species.

3. Discussion

The Discussion section centers on the genes identified as differentially expressed in this analysis, including those from both asexual and sexual stages of P. vivax. However, as in the Introduction, the authors fail to provide any substantive discussion or comparison with studies that have investigated rosetting in P. falciparum. This discussion is essential, as it would not only place the present findings in a broader scientific context but also highlight the importance of P. vivax–specific research.

4. Methods:

The Methods section is clearly written and provides sufficient detail to allow replication of the experiments. However, the rationale for the selection of the six samples included in the RNA-seq analysis is not explained. The authors should clarify the criteria used for sample selection.

5. Figures:

Figures 1 and 2 from the main text are clear and easy to understand. However, Figure 3 is confused and the legend inside the Pie chart makes it even harder to understand. I would recommend a figure with the 3 pie charts separated and with a color legend below the charts.

The supplementary material is well presented and provide important information for the readers.

Typos:

Line 166: While glycophorin C is known to be an important receptor in resetting (Lee et al. 2014), no ligands have yet been described. (rosetting)

### REVIEWER #3

Abstract - include the main info of the work which is comparison between the transcritptome profile of LR and HR patients. Quantitative numbers of differential expressed genes and what they are related to (membrane, kinase, etc) must be add to the text

Line 109 - Add the volume unit to blood in the sentence "platelet levels <150,000 platelets per of blood"

Fig 1 legend - Include the statistical test used to discriminate the significant difference between LR and HR patients, the p value that the symbol *** represents, and non-filled circles meaning.

Page 6 - Topic P. vivax low input cDNA synthesis, library preparation and sequencing

It is fundamental to state that only 6 of the 16 patients had samples selected for transcriptomics

Supplementary table 1 - Add a new column to highlight whether the samples selected for transcriptomics were derived from a LR or HR patient and what rosetting percentage each patient displayed.

It would be very interesting to highlight the expression differences for the P. vivax membrane genes in each patient group in a novel figure or an extension of Fig 3 through another heamap, a table, etc.

Discussion - since the expression differences for parasite membrane genes were not shown in the results or figures, but only in the supplementary material, it is not clear whether their expression patterns follow an expected rational… also, no complementary comparison was made between these P. vivax data to a P. falciparum counterpart.

Methods

Ethical Approval - Add FMT-HVD meaning (a tertiary care centre for infectious disease in Manaus, Amazonas State, Brazil).

Study Area, Subjects and Sample Collection - Remove FMT-HVD meaning.

## AUTHORS' RESPONSE TO THE REVIEWERS

Rebutall Letter to Reviewer Comments

We thank the reviewers for their valuable comments and suggestions on this manuscript. Below, we provide point-by-point responses to each of the raised points (highlighted in red). Please note that the suggested revisions have been carefully considered and incorporated into the updated version of the manuscript.

Reviewer: 1

"it would be helpfull for future interpretation of the data if the authors add a table with the information of % of each parasite stage, for each isolate, at the moment of sample processing for transcriptomics".

R: Thank you for your contribution. We agree that the information indicating the prevalence of each stage of the parasite, for each isolate, at the time of sample processing for transcriptomics is relevant to complement the other data presented. Therefore, we have included a table with this information in the results section (red text, page 5, lines 137-140). The table was included in Bourgard_etal_20251101_TrackedVersion file (red text, page 5, lines 58-64).

Reviewer: 2

1.Abstract: In the abstract, the authors report that one third of the genes differentially expressed in P. vivax rosetting parasites were "conserved within Plasmodium and of unknown function." This statement, however, lacks sufficient context. The total number of differentially expressed genes must be provided; without it, the proportion cited cannot be properly interpreted or considered scientifically meaningful.

R: Thank you for your concerns and suggestions. The requested information about the total number of differentially expressed genes has been included in the summary section (red text, page 2, lines 45-48).

2.Introduction: In the Introduction/Background section, the authors provide a comprehensive overview of transcriptomic studies conducted on P. vivax samples. However, they must also include a discussion of the molecular mechanisms of rosetting in P. falciparum, a parasite for which this phenomenon has been more extensively studied. Incorporating this information would provide essential context for readers, enabling a more meaningful comparison and a clearer understanding of the differences in pathogenic mechanisms across malaria-causing species.

R: Certainly, information on rosette formation in P. falciparum is extremely important for understanding and facilitates understanding of the adhesive phenotype in P. vivax. The information has been inserted in the introduction section (red text, page 3, line 73-89).

3.Discussion The Discussion section centers on the genes identified as differentially expressed in this analysis, including those from both asexual and sexual stages of P. vivax. However, as in the Introduction, the authors fail to provide any substantive discussion or comparison with studies that have investigated rosetting in P. falciparum. This discussion is essential, as it would not only place the present findings in a broader scientific context but also highlight the importance of P. vivax–specific research.

R: Again, we agree that discussing information about rosette formation in P. falciparum is relevant and facilitates understanding of the complex adhesive phenomenon. The information has been inserted in the discussion section (red text, page 7, line 194-214).

4.Methods: The Methods section is clearly written and provides sufficient detail to allow replication of the experiments. However, the rationale for the selection of the six samples included in the RNA-seq analysis is not explained. The authors should clarify the criteria used for sample selection.

R: We agree that this is valuable information that should be included. Work samples were selected based on the quality and quantity of samples after the extraction step, given the difficulty of obtaining quality materials for RNA-seq. For this reason, four patients with low rosette formation rates (10%) were selected. The information was inserted in the methods section (text in red, page 14, lines 408-412).

5.Figures: Figures 1 and 2 from the main text are clear and easy to understand. However, Figure 3 is confused and the legend inside the Pie chart makes it even harder to understand. I would recommend a figure with the 3 pie charts separated and with a color legend below the charts. The supplementary material is well presented and provide important information for the readers.

R: We agree with your comment and have reconfigured Figure 3 in a different layout to make it easier to understand (Bourgard_etal_20251101_figures file, page 4, lines 46-76).

Typos: Line 166: While glycophorin C is known to be an important receptor in resetting (Lee et al. 2014), no ligands have yet been described. (rosetting)

R: Thank you for your careful review. The typos have been corrected (text in red, page 4, line 99; page 7, line 192).

Reviewer: 3

1.Abstract Include the main info of the work which is comparison between the transcritptome profile of LR and HR patients. Quantitative numbers of differential expressed genes and what they are related to (membrane, kinase, etc) must be add to the text

R: Thank you for your carefully considered suggestions. We have entered the requested data in the summary section (text in red, page 2, line 45-57).

Line 109 - Add the volume unit to blood in the sentence "platelet levels <150,000 platelets per of blood"

R: Thank you for the correction. We have included the complete information in this new version (platelet levels <150,000 platelets per microliter of blood) in the result section (text in red, page 5, line 131).

Fig 1 legend - Include the statistical test used to discriminate the significant difference between LR and HR patients, the p value that the symbol *** represents, and non-filled circles meaning.

R: A Mann Whitney test (p < 0.001) was used for statistical analyses. The unfilled circles represent isolates that were not included in the RNA-seq analyses due to the quality of the material after RNA extraction. This information was included in Bourgard_etal_20251101_figures file (red text, page 1, lines 6-8).

Page 6 - Topic P. vivax low input cDNA synthesis, library preparation and sequencing: It is fundamental to state that only 6 of the 16 patients had samples selected for transcriptomics.

R: We agree with your observation. The information was included in the result section (text in red, page 5, line 146-147) to improve understanding of the article's rationale. Supplementary table 1 - Add a new column to highlight whether the samples selected for transcriptomics were derived from a LR or HR patient and what rosetting percentage each patient displayed.

R: The suggestion was included in Supplementary table S1 in Bourgard_SupplMaterial_20251101 file (text in red, page 3, line 26). It would be very interesting to highlight the expression differences for the P. vivax membrane genes in each patient group in a novel figure or an extension of Fig 3 through another heamap, a table, etc.

R: The most highly expressed genes for each patient group are listed in Supplementary Table S8, according to positive (HR) and negative (LR) results for Log2FC. In any case, we highlight this information in the text, in the results section (text in red, page 6, line 183-185). Discussion - since the expression differences for parasite membrane genes were not shown in the results or figures, but only in the supplementary material, it is not clear whether their expression patterns follow an expected rational… also, no complementary comparison was made between these P. vivax data to a P. falciparum counterpart.

R: Unfortunately, the literature on rosette formation based on transcriptomic data for Plasmodium falciparum is outdated and needs further study. We did not find any P. falciparum study that could aid in the discussion of the results found in this study for P. vivax. Methods Ethical Approval - Add FMT-HVD meaning (a tertiary care centre for infectious disease in Manaus, Amazonas State, Brazil). R: Thank you for your attention. The meanings were increased in the methods section (red text, page 11, line 318-319). Study Area, Subjects and Sample Collection - Remove FMT-HVD meaning. R: We perform meaning deletion.

## SECOND REVIEW ROUND

### REVIEWERS' COMMENTS

**REVIEWER #1**

No comments

**REVIEWER #2**

No comments.

**REVIEWER #3**

No comments.

