## [Reviewer Report · FIRST REVIEW ROUND - REVIEWERS COMMENTS]

## REVIEWER #1

This manuscript investigates the transcriptional differences between *P. vivax* isolates with low rosetting and *P. vivax* isolates with high rosetting.

The data presented is quite descriptive, since it does not point to a clear rosetting mechanism.

However, it will be usefull for the community to advance studies with this goal.

Prior to genetic material isolation, the parasites were cultured for maturation so the same stage of each isolate was analysed.

However, *P. vivax* maturation vary a lot, with some isolates progressing to late troph/schizonts quite fast, while other seem to arrest in earlier stages in vitro.

Also, some isolates when cultured seem to have a higher than expected preference for gametocytogenesis.

Therefore, it would be helpfull for future interpretation of the data if the authors add a table with the information of % of each parasite stage, for each isolate, at the moment of sample processing for transcriptomics.

**REVIEWER #2**

The manuscript addresses important aspects of the molecular mechanisms underlying the resetting of *P. vivax*–infected red blood cells, a topic that remains poorly characterized in *P. vivax* biology.

The study presents original findings, and the results are described in a clear and comprehensive manner, making the work accessible even to readers who are not specialists in transcriptomic analysis.

The paper is short but informative, with all transcriptomic analysis presented in tables and supplementary figures.

It´s the first work that uses transcriptomics to understand *P. vivax* resetting.

Even with a low number of samples/ libraries analyzed (6 libraires) it provides important new information to the field.

Overall, the manuscript represents a valuable contribution to the field.

I recommend acceptance for publication pending minor revisions, as outlined below.

1. Abstract:

In the abstract, the authors report that one third of the genes differentially expressed in *P. vivax* rosetting parasites were “conserved within *Plasmodium* and of unknown function.”

This statement, however, lacks sufficient context. The total number of differentially expressed genes must be provided;

without it, the proportion cited cannot be properly interpreted or considered scientifically meaningful.

2. Introduction

In the Introduction/Background section, the authors provide a comprehensive overview of transcriptomic studies conducted on *P. vivax* samples.

However, they must also include a discussion of the molecular mechanisms of rosetting in *P. falciparum*, a parasite for which this phenomenon has been more extensively studied.

Incorporating this information would provide essential context for readers, enabling a more meaningful comparison and a clearer understanding of the differences in pathogenic mechanisms across malaria-causing species.

3. Discussion

The Discussion section centers on the genes identified as differentially expressed in this analysis, including those from both asexual and sexual stages of *P. vivax*.

However, as in the Introduction, the authors fail to provide any substantive discussion or comparison with studies that have investigated rosetting in *P. falciparum*.

This discussion is essential, as it would not only place the present findings in a broader scientific context but also highlight the importance of *P. vivax*–specific research.

4. Methods:

The Methods section is clearly written and provides sufficient detail to allow replication of the experiments.

However, the rationale for the selection of the six samples included in the RNA-seq analysis is not explained.

The authors should clarify the criteria used for sample selection.

5. Figures:

Figures 1 and 2 from the main text are clear and easy to understand.

However, Figure 3 is confused and the legend inside the Pie chart makes it even harder to understand.

I would recommend a figure with the 3 pie charts separated and with a color legend below the charts.

The supplementary material is well presented and provide important information for the readers.

Typos:

Line 166: While glycophorin C is known to be an important receptor in resetting (Lee et al. 2014), no ligands have yet been described.

(rosetting)

**REVIEWER #3**

Abstract - include the main info of the work which is comparison between the transcritptome profile of LR and HR patients.

Quantitative numbers of differential expressed genes and what they are related to (membrane, kinase, etc) must be add to the text

Line 109 - Add the volume unit to blood in the sentence “platelet levels <150,000 platelets per of blood”

Fig 1 legend - Include the statistical test used to discriminate the significant difference between LR and HR patients, the p value that the symbol *** represents, and non-filled circles meaning.

Page 6 - Topic *P. vivax* low input cDNA synthesis, library preparation and sequencing

It is fundamental to state that only 6 of the 16 patients had samples selected for transcriptomics

Supplementary table 1 - Add a new column to highlight whether the samples selected for transcriptomics were derived from a LR or HR patient and what rosetting percentage each patient displayed.

It would be very interesting to highlight the expression differences for the *P. vivax* membrane genes in each patient group in a novel figure or an extension of Fig 3 through another heamap, a table, etc.

Discussion - since the expression differences for parasite membrane genes were not shown in the results or figures, but only in the supplementary material, it is not clear whether their expression patterns follow an expected rational… also, no complementary comparison was made between these *P. vivax* data to a *P. falciparum* counterpart.

Methods

Ethical Approval - Add FMT-HVD meaning (a tertiary care centre for infectious disease in Manaus, Amazonas State, Brazil).

Study Area, Subjects and Sample Collection - Remove FMT-HVD meaning.

## AUTHORS’ RESPONSE TO THE REVIEWERS

Rebutall Letter to Reviewer Comments

We thank the reviewers for their valuable comments and suggestions on this manuscript.

Below, we provide point-by-point responses to each of the raised points (highlighted in red).

Please note that the suggested revisions have been carefully considered and incorporated into the updated version of the manuscript.

**Reviewer: 1**

“it would be helpfull for future interpretation of the data if the authors add a table with the information of % of each parasite stage, for each isolate, at the moment of sample processing for transcriptomics”.

R: Thank you for your contribution. We agree that the information indicating the prevalence of each stage of the parasite, for each isolate, at the time of sample processing for transcriptomics is relevant to complement the other data presented.

Therefore, we have included a table with this information in the results section (red text, page 5, lines 137-140).

The table was included in Bourgard_etal_20251101_TrackedVersion file (red text, page 5, lines 58-64).

**Reviewer: 2**

1.Abstract: In the abstract, the authors report that one third of the genes differentially expressed in *P. vivax* rosetting parasites were “conserved within *Plasmodium* and of unknown function.”

This statement, however, lacks sufficient context. The total number of differentially expressed genes must be provided;

without it, the proportion cited cannot be properly interpreted or considered scientifically meaningful.

R: Thank you for your concerns and suggestions. The requested information about the total number of differentially expressed genes has been included in the summary section (red text, page 2, lines 45-48).

2.Introduction: In the Introduction/Background section, the authors provide a comprehensive overview of transcriptomic studies conducted on *P. vivax* samples.

However, they must also include a discussion of the molecular mechanisms of rosetting in *P. falciparum*, a parasite for which this phenomenon has been more extensively studied.

Incorporating this information would provide essential context for readers, enabling a more meaningful comparison and a clearer understanding of the differences in pathogenic mechanisms across malaria-causing species.

R: Certainly, information on rosette formation in *P. falciparum* is extremely important for understanding and facilitates understanding of the adhesive phenotype in *P. vivax*.

The information has been inserted in the introduction section (red text, page 3, line 73-89).

3.Discussion The Discussion section centers on the genes identified as differentially expressed in this analysis, including those from both asexual and sexual stages of *P. vivax*.

However, as in the Introduction, the authors fail to provide any substantive discussion or comparison with studies that have investigated rosetting in *P. falciparum*.

This discussion is essential, as it would not only place the present findings in a broader scientific context but also highlight the importance of *P. vivax*–specific research.

R: Again, we agree that discussing information about rosette formation in *P. falciparum* is relevant and facilitates understanding of the complex adhesive phenomenon.

The information has been inserted in the discussion section (red text, page 7, line 194-214).

4.Methods: The Methods section is clearly written and provides sufficient detail to allow replication of the experiments.

However, the rationale for the selection of the six samples included in the RNA-seq analysis is not explained.

The authors should clarify the criteria used for sample selection.

R: We agree that this is valuable information that should be included.

Work samples were selected based on the quality and quantity of samples after the extraction step, given the difficulty of obtaining quality materials for RNA-seq.

For this reason, four patients with low rosette formation rates (10%) were selected.

The information was inserted in the methods section (text in red, page 14, lines 408-412).

5.Figures: Figures 1 and 2 from the main text are clear and easy to understand.

However, Figure 3 is confused and the legend inside the Pie chart makes it even harder to understand.

I would recommend a figure with the 3 pie charts separated and with a color legend below the charts.

The supplementary material is well presented and provide important information for the readers.

R: We agree with your comment and have reconfigured Figure 3 in a different layout to make it easier to understand (Bourgard_etal_20251101_figures file, page 4, lines 46-76).

Typos: Line 166: While glycophorin C is known to be an important receptor in resetting (Lee et al. 2014), no ligands have yet been described.

(rosetting)

R: Thank you for your careful review. The typos have been corrected (text in red, page 4, line 99; page 7, line 192).

**Reviewer: 3**

1.Abstract Include the main info of the work which is comparison between the transcritptome profile of LR and HR patients.

Quantitative numbers of differential expressed genes and what they are related to (membrane, kinase, etc) must be add to the text

R: Thank you for your carefully considered suggestions. We have entered the requested data in the summary section (text in red, page 2, line 45-57).

Line 109 - Add the volume unit to blood in the sentence “platelet levels <150,000 platelets per of blood”

R: Thank you for the correction. We have included the complete information in this new version (platelet levels <150,000 platelets per microliter of blood) in the result section (text in red, page 5, line 131).

Fig 1 legend - Include the statistical test used to discriminate the significant difference between LR and HR patients, the p value that the symbol *** represents, and non-filled circles meaning.

R: A Mann Whitney test (p < 0.001) was used for statistical analyses.

The unfilled circles represent isolates that were not included in the RNA-seq analyses due to the quality of the material after RNA extraction.

This information was included in Bourgard_etal_20251101_figures file (red text, page 1, lines 6-8).

Page 6 - Topic *P. vivax* low input cDNA synthesis, library preparation and sequencing: It is fundamental to state that only 6 of the 16 patients had samples selected for transcriptomics.

R: We agree with your observation. The information was included in the result section (text in red, page 5, line 146-147) to improve understanding of the article’s rationale.

Supplementary table 1 - Add a new column to highlight whether the samples selected for transcriptomics were derived from a LR or HR patient and what rosetting percentage each patient displayed.

R: The suggestion was included in Supplementary table S1 in Bourgard_SupplMaterial_20251101 file (text in red, page 3, line 26).

It would be very interesting to highlight the expression differences for the *P. vivax* membrane genes in each patient group in a novel figure or an extension of Fig 3 through another heamap, a table, etc.

R: The most highly expressed genes for each patient group are listed in Supplementary Table S8, according to positive (HR) and negative (LR) results for Log2FC.

In any case, we highlight this information in the text, in the results section (text in red, page 6, line 183-185).

Discussion - since the expression differences for parasite membrane genes were not shown in the results or figures, but only in the supplementary material, it is not clear whether their expression patterns follow an expected rational… also, no complementary comparison was made between these *P. vivax* data to a *P. falciparum* counterpart.

R: Unfortunately, the literature on rosette formation based on transcriptomic data for *Plasmodium falciparum* is outdated and needs further study.

We did not find any *P. falciparum* study that could aid in the discussion of the results found in this study for *P. vivax*.

Methods Ethical Approval - Add FMT-HVD meaning (a tertiary care centre for infectious disease in Manaus, Amazonas State, Brazil).

R: Thank you for your attention. The meanings were increased in the methods section (red text, page 11, line 318-319).

Study Area, Subjects and Sample Collection - Remove FMT-HVD meaning. R: We perform meaning deletion.